# Clonally related, Notch-differentiated spinal neurons integrate into distinct circuits

Saul Bello-Rojas, Martha W Bagnall*

Department of Neuroscience, Washington University in St. Louis, St. Louis, United States

**Abstract** Shared lineage has diverse effects on patterns of neuronal connectivity. In mammalian cortex, excitatory sister neurons assemble into shared microcircuits. In *Drosophila*, in contrast, sister neurons with different levels of Notch expression (Notch$^{ON}$/Notch$^{OFF}$) develop distinct identities and diverge into separate circuits. Notch-differentiated sister neurons have been observed in vertebrate spinal cord and cerebellum, but whether they integrate into shared or distinct circuits remains unknown. Here, we evaluate how sister V2a (Notch$^{OFF}$)/V2b (Notch$^{ON}$) neurons in the zebrafish integrate into spinal circuits. Using an in vivo labeling approach, we identified pairs of sister V2a/b neurons born from individual Vsx1+ progenitors and observed that they have somata in close proximity to each other and similar axonal trajectories. However, paired whole-cell electrophysiology and optogenetics revealed that sister V2a/b neurons receive input from distinct presynaptic sources, do not communicate with each other, and connect to largely distinct targets. These results resemble the divergent connectivity in *Drosophila* and represent the first evidence of Notch-differentiated circuit integration in a vertebrate system.

## Editor's evaluation

This is an important article that describes the connectivity of sibling neurons in the zebrafish spinal cord, where one sibling receives Notch signaling (Notch-ON) and the other does not (Notch-OFF). They find that V2a and V2b siblings have different morphology, inputs, outputs, and fail to connect to each other; this provides new insight into the role of lineage in specifying neuronal connectivity. The experiments are convincing and the conclusions are supported by the data presented.

*For correspondence:
bagnall@wustl.edu

Competing interest: The authors declare that no competing interests exist.

## Introduction

How does shared lineage affect neuronal circuitry? Neurons arising from common progenitors are more likely to exhibit stereotypic patterns of connectivity, in two models from vertebrate and invertebrate systems. In mouse cortex, clonally related excitatory sister neurons preferentially form connections within a shared microcircuit (*Xu et al., 2014*; *Yu et al., 2009*). In contrast, clonally related sister neurons in *Drosophila* form distinct Notch$^{ON}$ and Notch$^{OFF}$ hemilineages which innervate distinct targets and often express different neurotransmitters (*Artavanis-Tsakonas et al., 1999*; *Endo et al., 2007*; *Harris et al., 2015*; *Lacin et al., 2019*; *Lacin and Truman, 2016*; *Mark et al., 2021*; *Pinto-Teixeira and Desplan, 2014*; *Skeath and Doe, 1998*).

Notch-differentiated clonally related sister neurons have been observed in the vertebrate spinal cord and cerebellum (*Kimura et al., 2008*; *Peng et al., 2007*; *Zhang et al., 2021*), but it remains unknown whether these clonally related neurons integrate into shared circuits. In ventral spinal cord, motor neurons and interneurons develop from five progenitor domains (p0, p1, p2, pMN, and p3)

**eLife digest** The brain is populated by neurons which are generated during embryonic development from cells called progenitors. Neurons that come from the same progenitor cell are considered to be 'sisters'. In certain brain regions of mice, sister neurons are often wired into shared networks, meaning they are more likely to receive input from the same neurons and connect with each other than non-sister cells.

In contrast, in invertebrate animals, like the fruit fly, sister neurons often have different identities and are less likely to connect with each other. This may be because sister neurons in fruit flies often have varied levels of a protein called Notch, which plays an important role in establishing the identity of cells. Vertebrate and invertebrate animals are different in many respects, and it remained unclear whether Notch levels dictate which sister neurons connect together in vertebrates as they do in fruit flies.

To investigate, Bello-Rojas and Bagnall studied two neurons in the spinal cord of zebrafish embryos which come from the same type of progenitor cell: the V2a neuron which has low levels of Notch, and the V2b neuron which has high levels of Notch. Fish, like humans, are vertebrates; however, their embryos are mostly transparent, making it easier to track how their neurons make connections during development using a microscope. This enabled Bello-Rojas and Bagnall to monitor whether V2a and V2b sister neurons joined the same network, like in other vertebrates, or different networks, akin to sister neurons in fruit flies which also have differing levels of Notch.

Bello-Rojas and Bagnall found that sister V2a and V2b neurons stayed close to one another and seemed to connect through similar paths. However, closer investigation revealed that the sister neurons did not receive input from the same source. They also did not connect to each other or the same output neuron, suggesting that V2a and V2b sister neurons are part of different networks.

This is the first time Notch levels have been shown to regulate which network a neuron will join in a vertebrate species. Since the V2a and V2b neurons are involved in controlling body movement, future work should determine whether adding progenitor cells that produce these neurons into the spinal cord could help the neuron network recover after injury or disease.

(*Goulding, 2009*; *Goulding and Lamar, 2000*; *Jessell, 2000*). Progenitors in the p2 domain transiently express the transcription factor Vsx1 (*Kimura et al., 2008*; *Passini et al., 1998*). Each p2 progenitor makes a final paired division into an excitatory V2a (Notch$^{OFF}$) and an inhibitory V2b (Notch$^{ON}$) neuron via asymmetric expression of Delta ligands and subsequent Notch-mediated lateral inhibition (*Del Barrio et al., 2007*; *Francius et al., 2016*; *Kimura et al., 2008*; *Okigawa et al., 2014*; *Peng et al., 2007*).

Although both V2a and V2b neurons project axons ipsilaterally and caudally, these neuron classes differ in other aspects. V2a interneurons express *vsx2* (referred to as *chx10* in this paper for clarity) and provide glutamatergic drive onto motor populations (*Kimura et al., 2006*), whereas V2b interneurons express *gata3* and provide glycinergic and GABAergic inhibition onto motor populations (*Andrzejczuk et al., 2018*; *Callahan et al., 2019*). V2b neurons also support flexor/extensor alternation through reciprocal inhibition in limb circuits (*Britz et al., 2015*; *Zhang et al., 2014*). Given their shared origin but divergent cellular identities, it remains unknown whether these V2a/b sister neurons integrate into shared or distinct functional spinal circuits.

We investigated whether V2a/b sister neurons in zebrafish spinal cord preferentially integrate into shared circuits, as with clonally related cortical neurons, or distinct circuits, as with Notch-differentiated hemilineages in *Drosophila*. Using a sparse labeling approach, we directly observed and identified individual pairs of sister V2a/b neurons arising from a single progenitor. Our morphological and electrophysiological analyses reveal that although sister V2a/b neurons share anatomical characteristics, these sister neurons diverge into separate circuits, with largely distinct presynaptic and postsynaptic partners. To the best of our knowledge, this is the first assessment of circuit integration of Notch-differentiated clonally related neurons in vertebrate models.

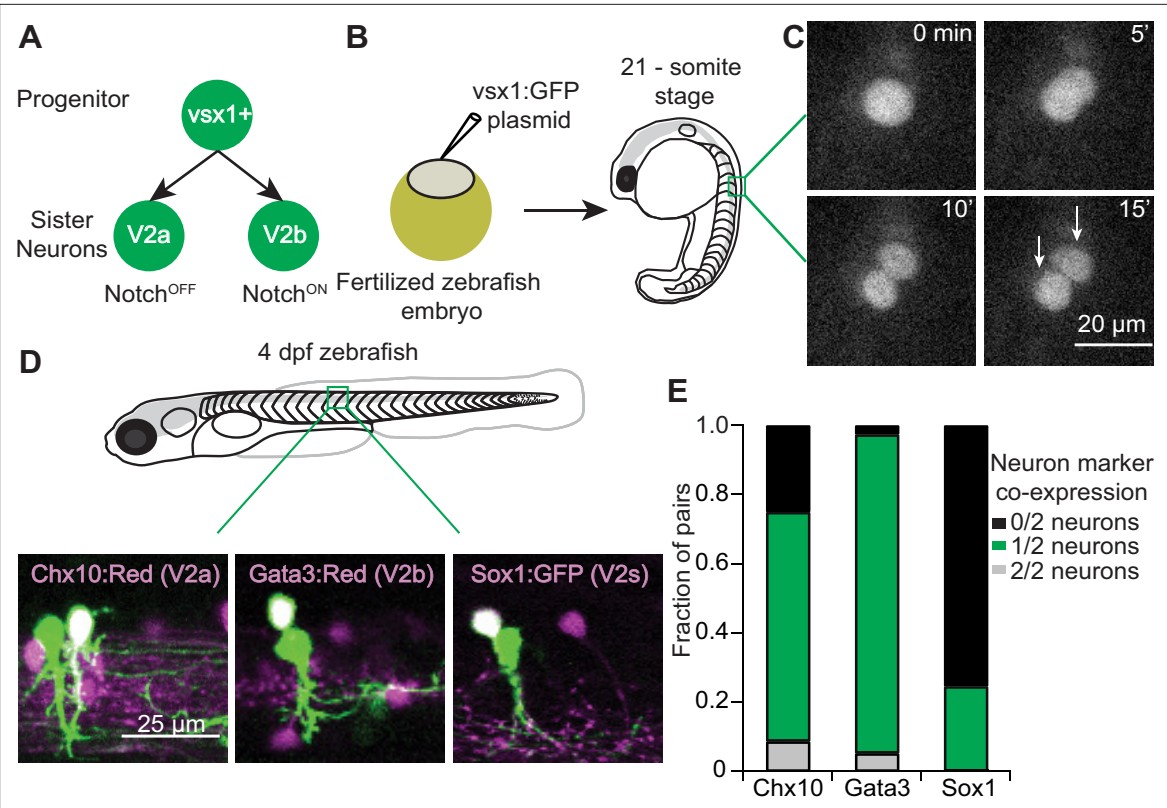

**Figure 1.** Sparse *vsx1+* progenitor labeling allows for clonal pair tracking in vivo. (**A**) Schematic of *vsx1* GFP+ progenitor undergoing a final paired division into sister V2a/b neurons. (**B**) Schematic of fertilized embryo injection and screening for *vsx1* GFP+ progenitors at the 21-somite stage. (**C**) Time-lapse single-plane confocal images taken every 5 min as a *vsx1* GFP+ progenitor divides into two sister neurons, imaged at 24 hr post fertilization (hpf). (**D**) Confocal imaging of *vsx1+* sister neuron pairs in the spinal cord of 4 dpf larvae. Left: *vsx1* GFP+ sister pair in a *chx10:Red* larva. One sister neuron is co-labeled (white, V2a) while the other is a presumed V2b. Middle: *vsx1* GFP+ sister pair in a *gata3:Red* larva showing an identified V2b with a presumed V2a or V2s. Right: *vsx1* mCherry+ sister pair in a *sox1:GFP* larva, showing an identified V2s with a presumed V2b. Colors switched for label and image consistency. (**E**) Bar graph displaying the fraction of *vsx1* GFP+ pairs in *chx10:Red* (n = 92), *gata3:Red* (n = 38), and *sox1:GFP* (n = 65) larvae in which either 0/2 sister neurons were co-labeled with the reporter (black), 1/2 sister neurons were co-labeled (green), or 2/2 sister neurons were co-labeled (gray). Source data for this figure are given in *Figure 1—source data 1*.

The online version of this article includes the following source data for figure 1:

**Source data 1.** Fraction of progenitors with reporter labels.

## Results

### Micro-injection of *vsx1* plasmid allows for clonal pair tracking in vivo

In both zebrafish and mice, *vsx1+* progenitors give rise to two distinct daughter populations, V2a and V2b neurons (*Kimura et al., 2008*; *Peng et al., 2007*). Using transgenic zebrafish, individual *vsx1+* progenitors have been shown to undergo a final paired division into one V2a (NotchOFF) and one V2b neuron (NotchON) (*Figure 1A*; *Kimura et al., 2008*). We aimed to develop a protocol to label and identify individual clonal pairs resulting from this division in vivo. To label individual pairs, we micro-injected titrated amounts of a bacterial artificial chromosome (BAC) construct, *vsx1:GFP*, into fertilized zebrafish embryos at the single-cell stage (*Figure 1B*). At the 21-somite stage, larval zebrafish were screened for *vsx1* GFP+ progenitors and then imaged every 5 min to capture the progenitor division (*Figure 1C*). Progenitors became elongated before dividing into two distinct cells.

When the fish become free swimming at 4 days post fertilization (dpf), *vsx1* GFP+ pairs were assessed for co-expression of known V2a/b transcription factors (*chx10/gata3*) to verify their neuronal identities, using transgenic fish Tg(*chx10:lox-dsRed-lox:GFP*) (*Kimura et al., 2006*) or Tg(*gata3:lox-dsRed-lox:GFP*) (*Callahan et al., 2019*; *Figure 1D*). For simplicity, these fish lines will be referred to as *chx10:Red* and *gata3:Red*. We assayed these in separate experiments due to overlap in fluorescence

from reporter lines. *Figure 1D* presents example images of *vsx1* GFP+ pairs in which one of the two neurons in the pair expresses the appropriate marker: a clonal pair (green) where one neuron co-expresses the V2a marker Chx10 (left), and a different clonal pair in which one neuron co-expresses the V2b marker Gata3 (middle). Based on previous work, we expected that every *vsx1* GFP+ pair would consist of one V2a and one V2b neuron (*Kimura et al., 2008*). However, among clonal pairs imaged in the *chx10:Red* background, only 61/92 (66.3%) of *vsx1* GFP+ pairs included one identified V2a neuron (*Figure 1E*). In contrast, in the *gata3:Red* background, 35/38 (92.1%) of *vsx1* GFP+ pairs included one identified V2b neuron (*Figure 1E*). Rarely, both *vsx1 GFP+* neurons in a pair expressed both Chx10 or Gata3 markers (<10%). However, in 25% of *vsx1* GFP+ pairs in *chx10:Red* fish, neither neuron expressed the V2a marker.

A possible explanation for the lower rate of V2a marker expression could be under-labeling in the fluorescent reporter line. Alternatively, there is at least one additional population of neurons to emerge from the V2 domain. In zebrafish, the V2s population is glycinergic and expresses Sox1a (*Gerber et al., 2019*). V2s neurons resemble V2c neurons in mice in that they both express Sox1a and arise after V2a/b development; however, V2c neurons are GABAergic while V2s neurons are purely glycinergic (*Gerber et al., 2019*; *Panayi et al., 2010*). Using the *Tg(sox1a:dmrt3a-gata2a:EFP(ka705))* reporter line, here referred to as *sox1:GFP*, (*Gerber et al., 2019*), we assayed the presence of *sox1+/vsx1+* neurons by injecting a *vsx1:mCherry* BAC in embryos at the single-cell stage. In 16/65 (24.6%) of *vsx1* mCherry+ pairs, one of the two sister neurons co-labeled with *sox1a* (*Figure 1D*, right), and in 49/65 (75.4%) of *vsx1* mCherry+ pairs, neither neuron co-labeled with *sox1a* (*Figure 1E*). These results suggest that not all *vsx1+* progenitors differentiate into V2a/b pairs. Instead, approximately 75% of *vsx1* progenitors divide into V2a/b pairs while the remainder divide into V2b/s pairs. We did not see any *vsx1+* triplets or singlets in co-label experiments (0/195), suggesting that *vsx1* progenitors only undergo a single, terminal division. Based on these results, we conclude that our stochastic labeling approach successfully labeled clonally related V2 neurons, but required a *chx10* co-label to properly identify *vsx1* pairs as V2a/b neurons in vivo.

## Sister V2a/b neurons remain in close proximity to each other

Immediately after progenitor division around 1 dpf, sister V2a/b neurons are located in close proximity to each other (*Kimura et al., 2008*), but they have not been followed out to 3–5 dpf when the spinal circuit transitions from spontaneous coiling during embryonic stages to the beat-and-glide locomotion at the larval stage. To assess somatic relationships between sister V2 neurons at larval stages, we measured inter-soma distances among sister and non-sister pairs. In an example fish (*Figure 2A*), the inter-soma distance between the GFP-only sister neuron (presumed V2b) to the GFP/Chx10:Red co-labeled V2a neuron (dark red arrow) was shorter than the distance between non-sister neurons (white arrows). Across animals, sister V2 neurons were usually closer neighbors than non-sister V2 neurons, whether measured in the V2a or V2b reporter lines (*Figure 2B*). Sister *vsx1+* neurons remained in close proximity to each other throughout embryonic and larval development. Beginning at 24 hpf, we embedded embryos in low melting point agarose, imaged, and then re-imaged at 48 hpf. At 24 hpf, sister neuron centers were ~7 µm apart, or effectively adjacent. By 48 hpf, this inter-soma distance increased slightly to ~9 µm (*Figure 2C*). In a separate set of experiments, we tracked *vsx1+* sister neurons from 48 to 96 hpf by embedding fish for imaging at 48 hpf, freeing from agarose after imaging, and re-embedding at 96 hpf. The distance between somata increased slightly, but still remained relatively short (*Figure 2C*). Because V2a/b somata are ~10 µm in size (*Callahan et al., 2019*; *Kimura et al., 2006*; *Menelaou et al., 2014*), our data suggest that sister V2 neurons usually remain adjacent. Lastly, restricting our analysis to sister V2a/b neurons using *chx10:Red* fish, we found that V2b neurons were typically positioned more dorsally than their sister V2a counterparts (*Figure 2D*), consistent with previous work showing inhibitory populations are located more dorsally than excitatory neurons in spinal cord (*Kimura et al., 2006*; *McLean et al., 2007*). Altogether, our data demonstrate that sister V2a/b neurons develop and remain close to each other during larval stages. As a result, in subsequent experiments we inferred that sparsely labeled *vsx1* GFP+ neurons located close to each other at 3–4 dpf represented sister pairs.

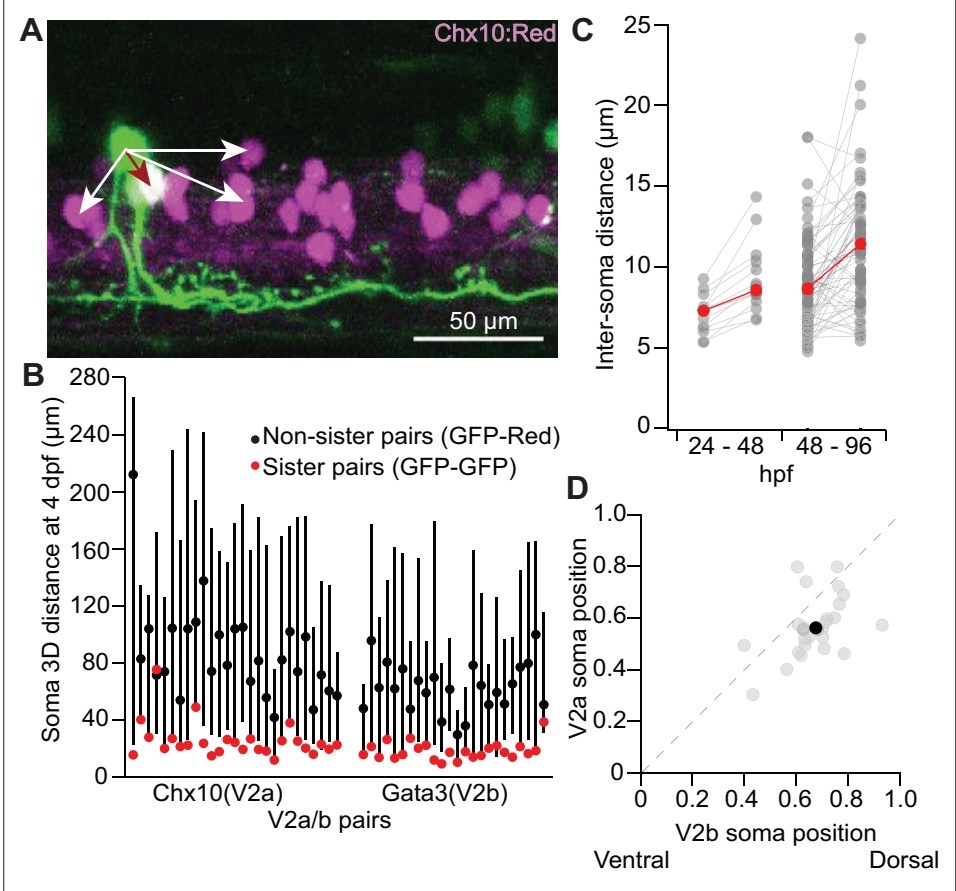

**Figure 2.** Sister V2a/b neurons remain in close proximity to each other. (**A**) Maximum intensity projection (50 planes, 50 µm) of *chx10:Red* with a single *vsx1* GFP+ clonal pair. The inter-soma distance between the GFP-only sister neuron to the GFP/Red co-labeled V2a neuron (dark red arrow) is smaller than the distance between non-sister neurons (white arrows). (**B**) For each clonal pair in either the Chx10 reporter line (n = 27) or the Gata3 reporter line (n = 24), the 3D distance between the two sister neurons (GFP-GFP, red) and the median 3D distance between one sister neuron and its non-sister neurons in the same segment (GFP-Red). Black dot indicates median and lines show 5th–95th percentiles. (**C**) Paired line plot of inter-soma distances of individual *vsx1* GFP+ sister pairs first imaged at 24 hr post fertilization (hpf) and later reimaged at 48 hpf (left) (n = 14) or first imaged at 48 hpf and later reimaged at 96 hpf (right) (n = 66). Red values indicate median distances at each time point. (**D**) Scatterplot of normalized dorsal (1)-ventral (0) soma position for each of 27 sister V2a/b neurons. Dashed line indicates unity. Typically, the V2b neuron was located more dorsally than the V2a neuron. Black dot indicates the median V2a/b pair position. **Wilcoxon signed-rank test, p=5.2 × 10⁻⁴, paired *t*-test, n = 27 pairs. Source data for this figure are given in *Figure 2—source data 1*.

The online version of this article includes the following source data for figure 2:

**Source data 1.** Intersoma distances and dorsal-ventral positions.

## Though V2a axons are consistently longer, sister V2a/b axons travel along similar trajectories

As V2a/b neurons both project descending, ipsilateral axons, we next assessed whether the axons of clonally related V2a/b neurons exhibited any consistent morphological characteristics. *Vsx1* GFP+ pairs were labeled in *chx10:Red* fish using a *vsx1*:GFP plasmid and later imaged on a confocal microscope. V2a/b axons were reconstructed (*Figure 3A*), and the descending axon length of each clonal V2a/b neuron was measured. Sister V2a neurons exhibited axons that were on average 61% longer than their V2b counterparts (*Figure 3B*), consistent with work showing that Notch expression attenuates axon growth (*Mark et al., 2021*; *Mizoguchi et al., 2020*). There was no relationship between the length of the axons and their location along the rostral-caudal axis of the fish (*Figure 3B*). To measure

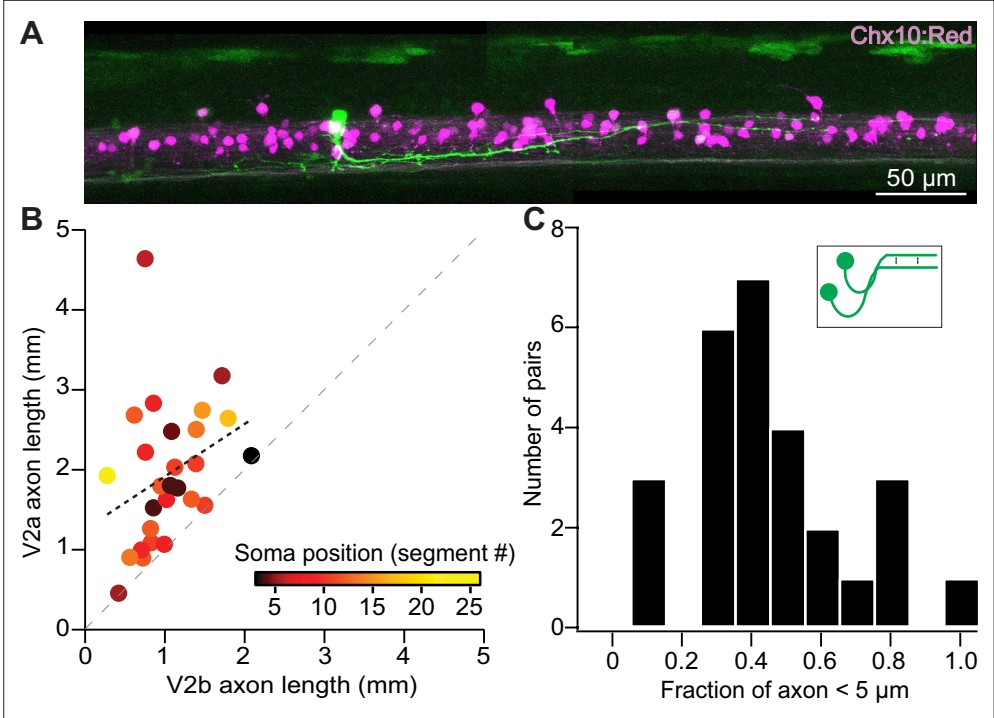

**Figure 3.** Sister V2a/b neurons project along similar trajectories, although V2a neurons are consistently longer. (**A**) Confocal image of *chx10:Red* larva exhibiting a single *vsx1* GFP+ clonal pair with long axons in close proximity to each other. Stitched maximum intensity projection over 74 z-planes (74 µm). (**B**) Scatterplot of sister V2a axon length vs. sister V2b axon length (n = 27) for V2a/b pairs. Heat map depicts the muscle segment number where each clonal pair was located. Black line depicts Pearson correlation, *r* = 0.32, p=0.10. V2a axons were invariably longer than sister V2b axons, as seen by each pair's position relative to the unity line (dashed gray). \*\*\*Wilcoxon signed-rank test, p=1.9 × 10$^{-5}$, paired *t*-test n = 27 pairs. (**C**) Histogram of clonal pairs showing the fraction of V2b axon that is within 5 µm of V2a axon. Inset schematic depicts how the distances were measured. Source data for this figure are given in *Figure 3—source data 1*.

The online version of this article includes the following source data for figure 3:

**Source data 1.** Axon lengths and fractions in close proximity.

axon proximity, the shortest distance between the V2b and V2a axon was calculated along each point of the V2b neuron, beginning at the axon hillock (*Figure 3C*, inset). The fraction of those inter-axon distances within 5 µm was calculated for each pair. Indeed, clonally related V2a/b neurons send axons along a similar trajectory, with a median of 37.1% of the V2b axon length within 5 µm of the V2a axon (*Figure 3C*). Because the axons follow similar paths, these results suggest a possibility for sister V2a/b neurons to contact shared synaptic targets.

## Sister V2a/b neurons receive input from distinct synaptic circuits

Work in hippocampus has shown that sister neurons are more likely to receive synaptic input from shared presynaptic partners than non-sister neurons (*Xu et al., 2014*). In contrast, Notch$^{ON}$ and Notch$^{OFF}$ sister neurons in *Drosophila* integrate into separate hemilineages that segregate spatially, although whether they receive shared input is not known (*Harris et al., 2015*). To evaluate whether sister V2a/b neurons receive input from shared or distinct presynaptic partners in vivo, we performed paired whole-cell electrophysiology in voltage clamp from clonally related pairs of V2a/b neurons identified as above (*Figure 4A and B*; *Bagnall and McLean, 2014*). Both sister neurons were held at −80 mV, the chloride reversal potential, to isolate excitatory postsynaptic currents (EPSCs), while a bright-field stimulus was used to elicit fictive swim (*Figure 4C*). The timing of EPSCs arriving in each neuron of the pair was asynchronous, as exemplified by an overlay of several hundred EPSCs from either a V2a/b and the associated EPSC-triggered average in its sister neuron (*Figure 4D and E*).

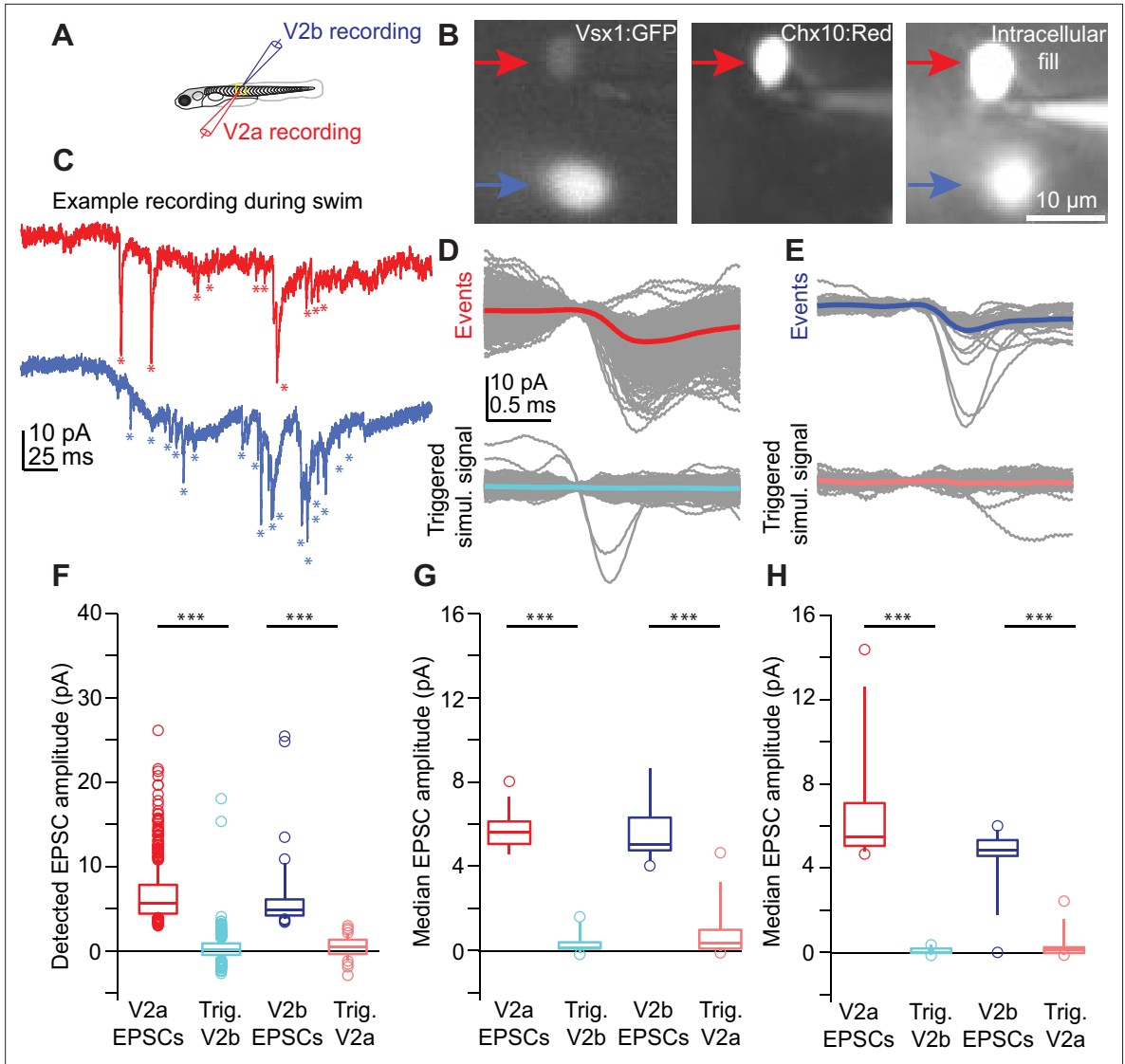

**Figure 4.** V2a/b sister neurons receive input from distinct synaptic circuits. (**A**) Schematic of larval zebrafish whole-cell paired recording (sister V2a in red and sister V2b in blue). (**B**) Two sister neurons labeled with *vsx1*:GFP (left), filled with dye during whole-cell recording (right). One neuron co-labels with V2a marker *chx10:Red* (middle). Red arrow and blue arrow indicate sister V2a and presumed sister V2b, respectively. (**C**) Example traces during swim of *vsx1* GFP+ sister neurons from V2a/b pair in voltage-clamp configuration. Asterisks denote detected excitatory postsynaptic current (EPSC) events. (**D**) Overlaid detected EPSC events recorded from sister V2a neuron (top) and simultaneously recorded signal in the sister V2b neuron (bottom). Most detected EPSCs in the V2a do not occur synchronously with EPSCs in the V2b neuron. (**E**) Overlaid detected EPSCs in V2b neuron (top) and simultaneously recorded signal in sister V2a neuron (bottom), also showing very few synchronous EPSCs. Colored traces represent averages of individual traces in gray. (**F**) Data from one example sister V2a/b pair showing the EPSC amplitude of detected events and the amplitude of the simultaneously recorded signal in the other neuron (Trig). Boxes depict medians, 25th and 75th percentiles. Whiskers denote 10th and 90th percentiles. Open circles depict EPSC values above and below the 10th and 90th percentiles. \*\*\*Wilcoxon signed-rank test (V2a – V2b simul.) p=1.8 × 10$^{-206}$; (V2b – V2a simul.) p=6.7 × 10$^{-15}$ paired *t*-test. (**G**) Summary data from all sister V2a/b pairs of recorded EPSC amplitudes and the EPSC-triggered simultaneously recorded signal in the other neuron. \*\*\*Wilcoxon signed-rank test, (V2a – V2b simul.) p=1.6 × 10$^{-5}$; (V2b – V2a simul.) p=2.6 × 10$^{-5}$ paired *t*-test, n = 13 pairs from 13 fish. (**H**) As in (**F**), for non-sister V2a/b paired recordings from the same spinal segment. \*\*\*Wilcoxon signed-rank test, (V2a – V2b simul.) p=1.6 × 10$^{-5}$; (V2b – V2a simul.) p=1.3 × 10$^{-4}$ paired *t*-test, n = 13 pairs from 13 fish. Source data for this figure are given in *Figure 4—source data 1*.

The online version of this article includes the following source data for figure 4:

**Source data 1.** Detected EPSC amplitudes.

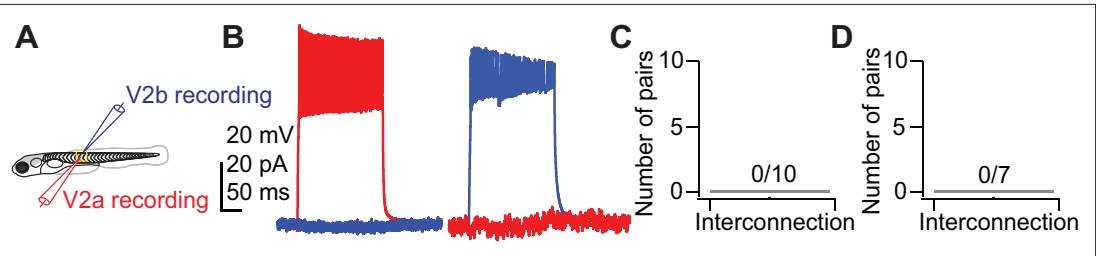

**Figure 5.** V2a/b sister neurons do not synapse with each other. (**A**) Schematic of larval zebrafish whole-cell paired recording. (**B**) Simultaneous current-clamp and voltage-clamp recording of sister V2a/b neurons. Current step-evoked spiking in sister V2a neuron and simultaneous voltage-clamp recording in V2b (left). Current step-evoked spiking in sister V2b neuron and simultaneous voltage-clamp recording in V2a (right). No synaptic responses are seen in either case. (**C**) Bar graph showing the number of clonal V2a/b interconnected pairs detected (n = 10 pairs from 10 fish). (**D**) Bar graph showing the number of V2a/b interconnected pairs detected for non-sister pairs (n = 7 pairs from seven fish).

A summary of the amplitudes of detected EPSCs and associated EPSC-triggered averages for this example neuron is shown in *Figure 4F*. Across recordings from 13 clonally related pairs in vivo, we consistently saw little to no synchronous synaptic input (*Figure 4G*).

Lastly, we wanted to compare whether this asynchrony in EPSC input was present in non-sister V2a/b neurons from the same segment. Using the same analysis, it appeared that non-sister V2a/b neurons from the same spinal segment receive input from distinct synaptic sources as well (*Figure 4H*). The asynchronous timing of these inputs suggests that they cannot be arriving from a shared presynaptic source, but rather, different presynaptic sources which fire at different times (*Bagnall and McLean, 2014*). Altogether, these data show that not only sister V2a/b neurons, but any V2a-V2b pair, clonal or non-clonal, from the same segment receives input from distinct presynaptic sources during light-evoked locomotion at slow to medium locomotor speeds. We cannot rule out the possibility that sister neurons receive shared inputs from circuits for fast locomotion or specialized behaviors.

## Sister V2a/b neurons do not form synaptic connections with each other

Clonal pair analysis in cortex has shown that sister neurons preferentially form synapses onto each other (*Yu et al., 2009*; *Zhang et al., 2017*). To identify whether sister V2a/b neurons form synaptic connections with each other, paired in vivo whole-cell recordings were performed in *chx10:Red* fish as described above (*Figure 5A*). Spiking was elicited by depolarizing current steps in either the V2a or the V2b neuron while the other neuron was held in voltage clamp to measure synaptic responses ($V_{hold}$ of –80 mV in V2b neurons to measure EPSCs, $V_{hold}$ of 0 mV in V2a neurons to measure IPSCs). In both cases, there were no detectable evoked currents, showing that sister V2a/b neurons do not connect with each other (*Figure 5C*). Similarly, non-clonally related V2a/b neurons exhibited no interconnectivity (*Figure 5D*). Therefore, V2a/b neurons in the same segment do not form direct synapses with each other. Any connectivity among V2a and V2b neurons likely occurs between neurons in different segments (*Sengupta and Bagnall, 2022*).

## Sister V2a/b neurons provide asymmetric input onto downstream neurons in spinal cord

Research in cortex has shown that clonally related inhibitory interneurons form synaptic connections with shared downstream targets (*Zhang et al., 2017*), although this claim is disputed (*Mayer et al., 2016*). Given the proximity of sister V2a/b axons (*Figure 3*), it was plausible that they share common downstream targets. To address this question, we micro-injected a *vsx1*:Gal4 BAC and a *UAS:CoChR2-tdTomato* plasmid (*Antinucci et al., 2020*; *Schild and Glauser, 2015*) in embryos at the single-cell stage to drive stochastic expression of this channelrhodopsin variant in *vsx1* sister neurons (*Figure 6A*) for selective optical stimulation of sister neurons. To distinguish presumed V2a/b pairs from V2b/s pairs, we screened *CoChR2-tdTomato+ vsx1* sister pairs for morphological characteristics ('Methods'). We validated that the optical stimuli effectively elicited spiking in *CoChR2-tdTomato+ vsx1* sister neurons by performing cell-attached recordings while providing a 10 ms light pulse (*Figure 6B*). All *CoChR2-tdTomato+ vsx1* neurons fired action potentials in response to optical stimulation (*Figure 6C*, n = 13 neurons from 12 fish). Spiking was elicited in both V2a and V2b *CoChR2-tdTomato+* neurons

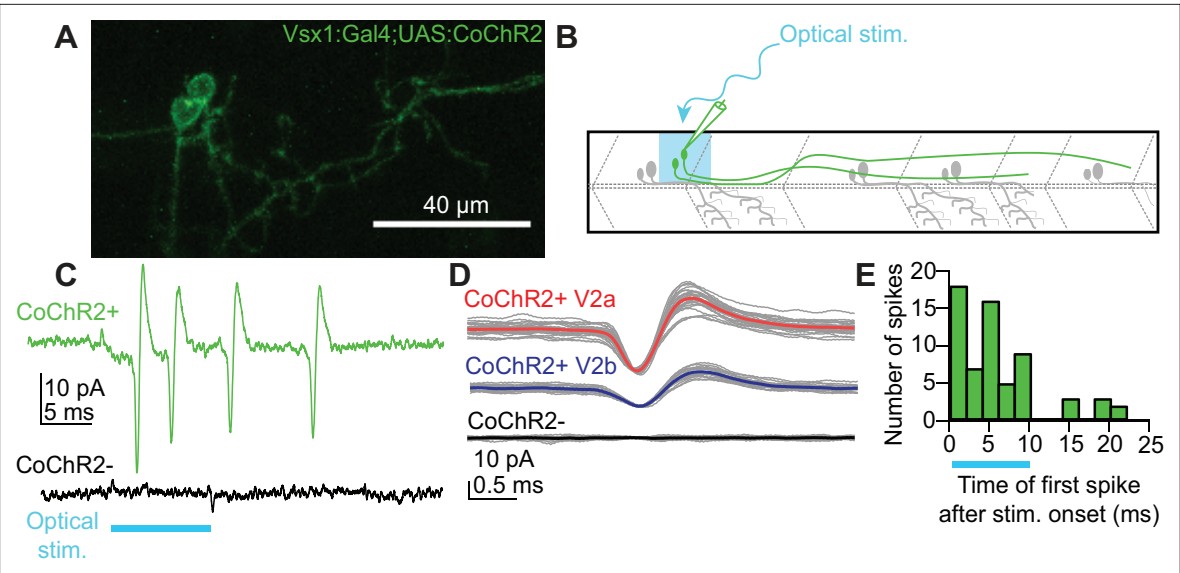

**Figure 6.** Optical stimulation elicits spiking in stochastically labeled *vsx1* sister neurons expressing CoChR2. (**A**) Maximum intensity projection (79 planes, 79 μm) of WT larva with a single *vsx1*:Gal4;UAS:*CoChR2-tdTomato+* clonal pair. (**B**) Schematic of cell-attached recording of *vsx1*:Gal4;UAS:*CoChR2-tdTomato+* neurons using optical stimulation. (**C**) Cell-attached example trace of *CoChR2*+V2a neuron during optical stimulation (top, green) (n = 13 from 12 fish). Cell-attached example trace of nearby *CoChR2-* neuron during optical stimulation (bottom, black) (n = 22 from 18 fish). Both example traces are aligned to the start of the 10ms optical stimulus (light blue). (**D**) Averaged detected spike events recorded from an example *CoChR2*+ sister V2a neuron (top, red), *CoChR2*+ sister V2b neuron (middle, blue), and absence of response in nearby *CoChR2-* neuron (bottom, black). (**E**) Histogram showing the number of spikes relative to the optical stimulus. Blue bar indicates the duration of the optical stimulus. Source data for this figure are given in *Figure 6—source data 1*.

The online version of this article includes the following source data for figure 6:

**Source data 1.** Spikes elicited by optical stimuli.

(*Figure 6D*). Similar experiments were performed on nearby *CochR2-tdTomato(-)* neurons to ensure that optical stimuli evoked spiking only in neurons expressing *CoChR2-tdTomato*. All *CoChR2-tdTomato(-)* neurons remained inactive during the light stimulus (*Figure 6C*, n = 22 neurons from 18 fish). In CoChR2-tdTomato+ neurons, most light-evoked spikes were observed throughout the duration of the stimulus, with some spiking following the stimulus window (*Figure 6C and E*). This prolonged activity is most likely due to the long inactivation kinetics of the *CoChR2* variant (*Antinucci et al., 2020*). Altogether, our optogenetic approach is a feasible method for assessing downstream connectivity of sister V2a/b neurons.

Having validated our optogenetic approach, we proceeded to perform whole-cell patch-clamp recordings on known V2a/b neuron downstream spinal targets (i.e., motor neurons, V1, V2a, V2b neurons), which were located 1–4 segments caudal to the V2a/b sister pair in voltage-clamp mode using a cesium-based internal solution (*Figure 7A*; *Bagnall and McLean, 2014*; *Callahan et al., 2019*; *Kimura et al., 2006*; *Menelaou and McLean, 2019*). Because sister *vsx1+* neurons are close to each other, our optogenetic stimulus would activate both neurons simultaneously. However, by clamping the target neuron at different reversal potentials, we could isolate either evoked EPSCs or inhibitory postsynaptic currents (IPSCs) (*Figure 7B–D*). In most recorded neurons, optical stimuli evoked neither EPSCs nor IPSCs, consistent with sparse connectivity in the spinal cord (n = 85/99; *Figure 7E and F*). In six target neurons, we recorded evoked EPSCs (V$_{hold}$ –80 mV) but not IPSCs, demonstrating that the target neuron received synaptic input from the CoChR2-labeled V2a neuron but not the V2b (*Figure 7B and E*). In another six target neurons, we detected evoked IPSCs (V$_{hold}$ 0 mV) but no EPSCs, demonstrating that the target neuron received synaptic input from the V2b but not the V2a neuron (*Figure 7C and E*). In a subset of experiments, NBQX/APV or strychnine were used to block responses and confirm glutamatergic or glycinergic connections, respectively (*Figure 7B*, n = 2; *Figure 7C*, n = 4). In two instances, a target neuron received both evoked EPSCs and IPSCs, with the magnitude

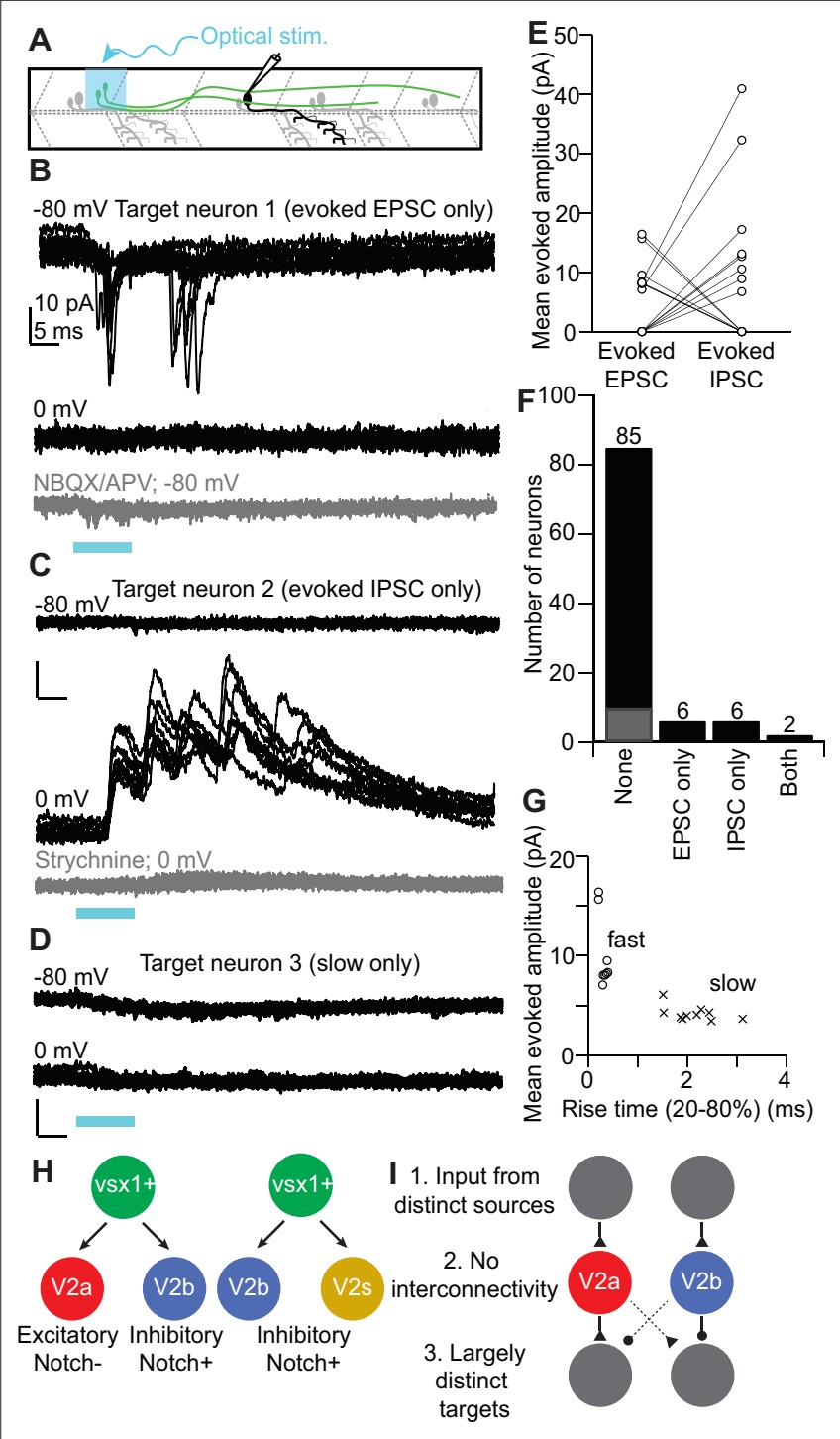

**Figure 7.** Sister V2a/b neurons provide asymmetric input onto downstream neurons in spinal cord. (**A**) Schematic of whole-cell recording of downstream neuronal targets of *vsx1*:Gal4;UAS:*CoChR2-tdTomato+* neurons using optical stimulation. (**B**) Example voltage-clamp traces from a target neuron held at $V_{hold}$ –80 mV or 0 mV during optical stimulation of the upstream sister neuron pair. Optical stimulation evoked excitatory postsynaptic currents (EPSCs) onto the target neuron (top) but not inhibitory postsynaptic currents (IPSCs) (middle), indicating connectivity from the V2a but not the V2b. Bottom: application of glutamatergic antagonists blocks the evoked EPSCs. (**C**) As in (**B**) for another target neuron, this one showing evoked IPSCs but not EPSCs. IPSCs were abolished by application of strychnine (bottom). (**D**) Example voltage-clamp traces from a target neuron held at $V_{hold}$ –80 mV or 0 mV. Trace showing a small, slow evoked EPSCs without any fast component. These are

*Figure 7 continued on next page*

*Figure 7 continued*

presumably due to indirect (polysynaptic) electrical connectivity from the optogenetically activated V2a neuron. (**E**) Mean evoked amplitude of optogenetically evoked EPSCs and IPSCs in each target neuron. 12/14 synaptically connected targets received only EPSCs or IPSCs, while 2/14 neurons received both EPSCs and IPSCs. (**F**) Bar graph depicting the number of EPSC only (n = 6), IPSC only (n = 6), both EPSC/IPSC (n = 2), and no responses (n = 75) (black) or only slow presumed polysynaptic (gray) (n = 10) detected across all target neurons recorded. (**G**) Scatterplot showing the distinction between mean evoked amplitude and 20–80% rise time for fast and slow evoked EPSC responses. (**H**) Schematic depicting the two presumed types of *vsx1* GFP+ sister pairs observed. (**I**) Summary of circuit integration pattern observed among sister V2a/b pairs. Source data for this figure are given in *Figure 7—source data 1*.

The online version of this article includes the following source data for figure 7:

**Source data 1.** Mean evoked synaptic amplitudes and rise times.

of IPSCs approximately fivefold larger than the magnitude of the EPSCs, suggesting an asymmetric connection from sister V2a/b neurons (*Figure 7E*).

In 10 neurons, we detected a slow depolarizing current when target neurons were held at –80 mV (*Figure 7D and F*, gray), but not at 0 mV. This evoked current had a lower amplitude and longer rise time than fast evoked EPSCs (*Figure 7G*). This slow excitatory current may be caused by a weak disynaptic electrical connection (*Menelaou and McLean, 2019*), but we were not able to eliminate it with gap junction blockers (carbenoxylone and 18-β-glycyrrhetinic acid). We summarize the identities of target neurons receiving synaptic input from sister V2a/b neurons in *Table 1*. Target neurons were evenly divided between motor neurons (early and late born), and excitatory and inhibitory interneurons. Overall, these results demonstrate that clonally related V2a/b neurons do not preferentially form synaptic connections with shared targets.

## Discussion

In this study, we showed that clonally related V2a/b neurons exhibit similar morphological characteristics, but form synapses with and receive information from largely distinct neuronal partners. Through our use of plasmid injections and time-lapse imaging, we definitively identified individual pairs of clonally related V2a/b neurons born from a single *vsx1+* progenitor cell in vivo (*Figure 7H*). Additionally, some *vsx1+* progenitors appear to divide into V2b/s pairs. Within V2a/b pairs, we saw that sister neuron somata remain in close proximity to each other and send their axons along similar trajectories. However, our electrophysiological data showed that these sister neurons integrate into distinct circuits. Clonally related V2a/b neurons do not communicate with each other, do not receive input from similar sources, and infrequently connect to the same downstream target. This connectivity pattern resembles circuitry seen in *Drosophila* Notch-differentiated hemilineages (*Figure 7I*). Our results represent the first evidence of Notch-differentiated circuit integration in a vertebrate system, and may reflect a means of cell-type and circuit diversification in earlier evolved neural structures.

**Table 1.** Sister V2a/b identified postsynaptic targets.

| Target type | EPSC only | IPSC only | Both | Total |
|---|---|---|---|---|
| Primary motor neuron | 1 | 1 | 1 | 3 |
| Secondary motor neuron | 2 | 1 | 0 | 3 |
| V1 | 0 | 2 | 0 | 2 |
| V2a | 0 | 2 | 0 | 2 |
| V2b | 1 | 0 | 0 | 1 |
| Unidentified | 2 | 0 | 1 | 3 |

EPSC, excitatory postsynaptic current; IPSC, inhibitory postsynaptic current.

## Notch determines cellular identity of *vsx1+* sister neurons

Notch is an important regulator in V2a/b differentiation, and during *vsx1+* progenitor division, differences in Notch expression result in the onset of V2a (Notch$^{OFF}$) or V2b (Notch$^{ON}$) programming (*Batista et al., 2008*; *Debrulle et al., 2020*; *Kimura et al., 2008*; *Mizoguchi et al., 2020*; *Okigawa et al., 2014*). However, it remains unknown whether Notch plays a role in sister V2a/b development beyond initiating cellular identity or whether it functions as an intermediary step before other molecular factors determine cellular morphology postmitotically (*Kozak et al., 2020*; *Mizoguchi et al., 2020*). Our morphological analysis showed differences in V2a/b axon lengths and dorsoventral position (*Figures 2 and 3*). Further experiments are needed to evaluate whether these differences are a result of Notch signaling or intrinsic to postmitotic cellular identity. Constitutive manipulation of Notch levels results in a skewing of V2a/b numbers (*Mizoguchi et al., 2020*). Therefore, a temporally controlled manipulation is needed to address the role of Notch in postmitotic morphological and functional development of sister V2a/b neurons.

Similarly, the recently discovered V2s population relies on Notch signaling for its development with Notch KO mutants showing a decrease in *sox1a+* neurons (*Gerber et al., 2019*). We speculate that some *vsx1+* progenitors give rise to V2b/s sister pairs in addition to the previously described V2a/b pairs. Our experiments in reporter lines (*Figure 1*) suggested that ~75% of *vsx1* GFP+ progenitors divided into V2a/b sister neurons, whereas ~25% resulted in V2b/s neuron pairs. V2s neurons arise later than the initial wave of V2a/b pairs (*Gerber et al., 2019*). Because Notch has been shown to exhibit different effects on cellular identity during early and late development, we suggest that delayed Notch activity causes some later born *vsx1+* sister neurons to adopt a V2b/s identity, which are both Notch$^{ON}$ (*Jacobs et al., 2022*). Similarly, only early cerebellar progenitors appear to undergo Notch differentiation into distinct cell types, the Purkinje and granule cells (*Zhang et al., 2021*). Notch overexpression experiments could have biased differentiation in favor of V2b/s pairs earlier in development, accounting for the increase in V2b and decrease in V2a numbers (*Mizoguchi et al., 2020*). However, these experiments have not looked at changes to V2s numbers, so selective evaluation of later born V2 progenitors is needed to identify whether V2b/s clonal pairs exist and if so whether they are temporally delayed relative to V2a/b pairs.

## Notch-mediated development influences circuit formation

Lineage pathfinding and innervation differences in *Drosophila* are well documented, and Notch-differentiated sister neurons in these organisms develop different axon trajectories, presumably connecting to different downstream targets (*Harris et al., 2015*; *Truman et al., 2010*). Our data show that *vsx1+* sister neurons in spinal cord have similar descending trajectories albeit different axon lengths (*Figure 3*). Analysis of Notch-differentiated lineages in vertebrate cerebellum have shown that Notch mediates cerebellar progenitor differentiation into excitatory and inhibitory cerebellar cell types (*Zhang et al., 2021*), but it is not yet known whether the resulting neurons integrate into shared or distinct circuits. Our results are consistent with a framework in which the progeny of Notch-differentiated divisions preferentially integrate into distinct networks in both invertebrates and vertebrates. In contrast with cortical lineages, the divergent cellular identities of sister V2a/b neurons appear to determine that they participate in distinct circuits. We speculate that earlier evolved neural structures rely on Notch-differentiated divisions as a means to diversify neuronal populations during development. The presence of Notch-differentiated sister neurons in both cerebellum and spinal cord could represent an efficient mechanism to generate diverse cell types early in development, in contrast to cortical reliance on dedicated streams of excitatory and inhibitory neural progenitors (*Goulding, 2009*; *Leto et al., 2016*; *Ma et al., 2018*). This would allow for the development of several neuronal cell types, each governed by their own intrinsic molecular cues.

## Shared *vsx1+* progenitor birthdates do not lead to shared integration

Developmental timing allows for proper integration of neurons into functional speed-dependent locomotor circuits. In zebrafish, motor neurons and interneurons born during similar developmental windows are active and recruited at similar speeds (*McLean et al., 2007*; *McLean and Fetcho, 2009*). These birthdate-dependent microcircuits emerge in larvae and persist into adulthood (*Ampatzis et al., 2014*). By definition, *vsx1+* sister neurons share a birthdate, suggesting that both neurons are likely recruited at similar speeds and therefore might integrate into shared microcircuits. However, our work

shows that *vsx1* sister neurons neither synapse onto each other, receive similar inputs, nor frequently target the same neurons. One possible explanation is that sister V2a/b divergence in cellular identity may cause integration into different hemilineage temporal cohorts, similar to *Drosophila*, which then determine their neuronal connectivity (*Mark et al., 2021*). Additionally, V2b neurons, whose recruitment patterns have not yet been described, may participate in different behaviors than V2a neurons. This separation of pathways driving excitatory and inhibitory neurons would allow for independent activation (accelerator) or inactivation (brake) of movement (*Callahan et al., 2019*; *Eklöf-Ljunggren et al., 2012*). It is worth noting that we measured synaptic inputs during fictive locomotion induced by bright-field stimuli, and that the possibility remains sister *vsx1* neurons do receive similar inputs under different behavioral paradigms, such as turns or escapes.

Lastly, the sister *vsx1* neurons infrequently connected to the same downstream targets (*Figure 7*). Because we saw two examples of targets receiving input from both the V2a and V2b neuron of a clonal pair, it is unclear whether sister neurons are explicitly discouraged from sharing downstream targets or whether it is simply random. In either case, the observed connectivity divergence might function to coordinate antagonistic components during locomotion. Spinal V1 interneurons target different populations of neurons along the rostral-caudal length of the spinal cord (*Sengupta et al., 2021*). Even if non-clonally related V2a and V2b neurons generally form synaptic contacts onto the same populations, such as motor neurons, they may exhibit different connectivity patterns in the longitudinal axis, preventing clonally related pairs from sharing downstream targets. Mapping the rostrocaudal connectivity of V2a and V2b populations would address this hypothesis.

# Materials and methods

## Key resources table

| Reagent type (species) or resource | Designation | Source or reference | Identifiers | Additional information |
|---|---|---|---|---|
| Genetic reagent (*Danio rerio*) | chx10:loxP-dsRed-loxP:GFP (synonym: Tg(vsx2:LOXP-DsRed-LOXP-GFP)) | *Kimura et al., 2006* | ZDB-ALT-061204-4 | BAC line generation |
| Genetic reagent (*D. rerio*) | gata3:loxP-dsRed-loxP:GFP (synonym: TgBAC(gata3: LOXP-DsRed-LOXP-GFP)) | *Callahan et al., 2019* | ZDB-ALT-190724-4 | BAC line generation |
| Genetic reagent (*D. rerio*) | sox1a: EGFP(ka705) (synonym: Tg(nccr. dmrt3a-gata2a:EGFP)) | *Gerber et al., 2019* | ZDB-ALT-191113-2 | |
| Genetic reagent (*D. rerio*) | chx10:GFP (synonym: TgBAC(vsx2:EGFP)) | *Kimura et al., 2006* | ZDB-ALT-061204-4 | BAC line generation |
| Recombinant DNA reagent | UAS:CoChR2-tdTomato | *Antinucci et al., 2020* | Addgene Cat# 124233 | Tol2 Plasmid |
| Recombinant DNA reagent | vsx1:GFP Plasmid | *Kimura et al., 2008* | | BAC CH211-67N1 |
| Recombinant DNA reagent | vsx1:mCherry Plasmid | This paper | | BAC CH211-67N1 |
| Recombinant DNA reagent | vsx1:Gal4 Plasmid | This paper | | BAC CH211-67N1 |
| Software, algorithm | Imaris 9.8 | Bitplane | https://imaris.oxinst.com/versions/9-8 | |
| Software, algorithm | ImageJ 1.53q | FIJI; *Schindelin et al., 2012* | | |
| Software, algorithm | Igor Pro 6.37 | Wavemetrics; *Rothman and Silver, 2018* | https://github.com/bagnall-lab/Event-detection; *Liu et al., 2020* | |

## Experimental model and subject details

All fish used for experiments were 0 to 6 dpf, before the onset of sexual maturation. All experiments and procedures were approved by the Animal Studies Committee at Washington University and adhere to NIH guidelines. Adult zebrafish (*Danio rerio*) were maintained at 28.5°C with a 14:10 light:dark cycle in the Washington University Zebrafish Facility up to 1 year following standard care

procedures. Larval zebrafish used for experiments were kept in Petri dishes in system water or housed with system water flow.

To target V2a and V2b neurons, the *Tg(chx10:loxP-dsRed-loxP:GFP)* (*Kimura et al., 2006*) (ZDB-ALT-061204-4) and *Tg(gata3:loxP-dsRed-loxP:GFP)* (*Callahan et al., 2019*) (ZDB-ALT-190724-4) lines were used. We visualized V2s neurons in *Tg(sox1a:dmrt3a-gata2a:EFP(ka705))* (ZDB-ALT-191113-2) (*Gerber et al., 2019*), a gift from Dr. Uwe Strähle.

## Stochastic single-cell labeling by microinjections

*Tg(chx10:loxP-dsRed-loxP:GFP)* and *Tg(gata3:loxP-dsRed-loxP:GFP)* were injected with a *vsx1:GFP* bacterial artificial chromosome (BAC) at a final concentration of 5 ng/µL (a gift from Dr. Shin-ichi Higashijima). *Tg(sox1a:dmrt3a-gata2a:EFP(ka705))* were injected with a *vsx1:mCherry* BAC at 15 ng/µL (generated from BAC CH211-67N1 by VectorBuilder, Inc). To label clonal pairs with an optogenetic activator, wild-type embryos were injected with a *vsx1:Gal4* BAC (VectorBuilder) and *UAS:CoChR2-tdTomato* plasmid (Addgene Cat# 124233) at 20 ng/µL and 25 ng/µL, respectively. The embryos were transferred to system water to develop. Embryos were screened between 1 and 4 dpf for sparse expression of Red/GFP fluorophores and selected for confocal imaging and electrophysiology. In optogenetic experiments, it was not feasible to use *Tg(chx10:Red)* animals to identify V2a neurons because of fluorophore overlap. Instead, we screened *CoChR2-tdTomato+ vsx1* sister neurons for distinguishable V2a morphology, specifically the presence of an ascending collateral, which is characteristic of V2a but not V2b or V2s neurons (*Callahan et al., 2019*; *Gerber et al., 2019*; *Menelaou et al., 2014*).

## Confocal imaging

At 18–24 hpf, larvae were anesthetized in 0.02% MS-222 and embedded in low-melting point agarose (0.7%) in a 10 mm FluoroDish (WPI). Spinal segments with sparsely labeled progenitors were imaged with a time-lapse approach, consisting of one Z-stack every 5 min, under a spinning disk confocal microscope (Crest X-Light V2; laser line 470 nm; upright Scientifica microscope; 40× objective; imaged with Photometrics BSI Prime camera). After progenitor division, larvae were kept in the FluoroDish inside of an incubator and reimaged at a higher-resolution at 2 dpf with a laser confocal (Olympus FV1200, 488 nm laser, XLUMPlanFl-20× W/0.95 NA water immersion objective).

Larvae imaged beginning at 2 dpf were anesthetized in 0.02% MS-222 and embedded in low melting point agarose (1.5%) in a 10 mm FluoroDish (WPI). Images were acquired on an Olympus FV1200 confocal microscope as above. A transmitted light image was obtained along with laser scanning fluorescent images to identify spinal segments. Sequential scanning used for multi-wavelength images. Fish were unembedded from the agarose and placed separately in labeled Petri dishes and later reimaged at 4 dpf as described above. In some cases, fish were only imaged at 4 dpf using the embedding methods described above. Transcription factor co-expression was evaluated visually.

## Image analysis

Confocal images were analyzed using Imaris (9.8, Bitplane) and ImageJ (1.53q, FIJI) (*Schindelin et al., 2012*). For axon tracing, stitched projection images were made with the Pairwise stitching (*Preibisch et al., 2009*) ImageJ plugin. The overlap of the fused image was smoothed with linear blending and was registered based on the fill channel or the average of all channels. Three-dimensional (3D) images were reconstructed and analyzed using Imaris. Axon length measurements of each reconstructed neuron were obtained using the Filament function to trace over the 3D rendering. Axon length includes only the descending branches of the neuron, starting at the axon hillock. 3D axon coordinates of descending projections were exported from Imaris, and separation of axon distances was calculated as the shortest distance between sister V2b to sister V2a axons. Muscle segment number was counted under differential interference contrast (DIC). Distances between soma centers were measured in three dimensions using the Points function in Imaris. Normalized dorsoventral soma position was calculated by measuring the distance of the soma from the notochord and dividing by the total height of the spinal cord, with 0 as the ventralmost point.

## Electrophysiological recordings

Cell-attached recordings were targeted to stochastically labeled WT fish with *vsx1:Gal4* BAC and *UAS:CoChR2-tdTomato* plasmids to calibrate firing of *vsx1:Gal4;UAS:CoChR2-tdTomato vsx1+* pairs. Whole-cell patch-clamp recordings were performed in *Tg(chx10:loxP-dsRed-loxP:GFP)* injected with *vsx1:GFP* and *Tg(chx10:GFP;gata3:loxP-dsRed-loxP:GFP)* larvae at 4–6 dpf for paired clonal V2a/b and non-clonal V2a/b recordings, respectively. Additionally, whole-cell patch-clamp recordings were performed in stochastically labeled *WT* fish with *vsx1:Gal4* BAC and *UAS:CoChR2-tdTomato* in downstream targets. Larvae were immobilized with 0.1% α-bungarotoxin and fixed to a Sylgard lined Petri dish with custom-sharpened tungsten pins. Each larva was then transferred to a microscope (Scientifica SliceScope Pro) equipped with infrared differential interface contrast optics, epifluorescence, and immersion objectives (Olympus: 40×, 0.8 NA). One muscle segment overlaying the spinal cord was removed (segments 7–17) using a blunt-end glass electrode and suction (**Wen and Brehm, 2010**). The bath solution consisted of (in mM) 134 NaCl, 2.9 KCl, 1.2 $MgCl_2$, 10 HEPES, 10 glucose, and 2.1 $CaCl_2$. Osmolarity was adjusted to ~295 mOsm and pH to 7.5.

Patch pipettes (5–15 MΩ) were filled with internal solution for voltage and current clamp and cell-attached composed of (in mM) 125 K gluconate, 2 $MgCl_2$, 4 KCl, 10 HEPES, 10 EGTA, and 4 $Na_2ATP$. Whole-cell optogenetic and some paired recordings were performed using internal solution composed of (in mM) 122 cesium methanesulfonate, 1 tetraethylammonium-Cl, 3 $MgCl_2$, 1 QX-314 Cl, 10 HEPES, 10 EGTA, and 4 $Na_2ATP$. Additionally, Alexa Fluor 647 hydrazide 0.05–0.1 mM or sulforhodamine (0.02%) was included to visualize morphology of recorded cells post hoc. Osmolarity was adjusted to ~285 mOsm and KOH or CsOH was used to bring the pH to 7.5. Patch recordings were made in whole-cell configuration using a Multiclamp 700B, filtered at 10 kHz (current clamp) or 2 kHz (voltage clamp). All recordings were digitized at 100 kHz with a Digidata 1440 (Molecular Devices) and acquired with pClamp 10 (Molecular Devices). The following drugs were bath applied where noted: strychnine (10 μM), NBQX (10 μM), APV (100 μM), 18-beta-glycyrrhetinic acid (150 μM), and carbenoxolone disodium salt (500 μM).

During paired electrophysiology recordings, fictive swimming sometimes occurred spontaneously and in other instances was elicited by white light illumination of the animal. In optogenetic experiments examining channelrhodopsin firing and V2a/b targeting, light stimulation was provided with high-intensity epifluorescent illumination (CoolLED pE-300), 10% intensity with a 40× (0.8 NA) water immersion objective for 10 ms. The objective was positioned over a single spinal segment prior to stimulus delivery.

Electrophysiology data were imported into Igor Pro 6.37 (Wavemetrics) using NeuroMatic (**Rothman and Silver, 2018**). The detection algorithm was based on the event detection instantiated in the SpAcAn environment for Igor Pro (**Rousseau et al., 2012**) and as previously described (**Bagnall and McLean, 2014**). All events detected were additionally screened manually to exclude spurious noise artifacts. EPSCs were analyzed using custom-written codes in Igor and MATLAB. For synchronized input evaluation, recordings were excluded from analysis if we could not induce robust fictive locomotion.

## Statistics

Statistical tests were performed using MATLAB (R2018a, MathWorks). Due to the non-normal distribution of physiological results, we used nonparametric statistics and tests for representations and comparisons. Details of statistical tests, p-values used, and sample sizes are described in the corresponding figure legends.

## Acknowledgements

We thank Dr. Rebecca Callahan for initial contribution in experimental design and topic development, Dr. Mohini Sengupta for thoughtful critiques of the paper, Dr. Shin-ichi Higashijima for the *vsx1:GFP* BAC construct, and Dr. Uwe Strähle for the *sox1:GFP* fish line. We are grateful to Drs. Andreas Burkhalter and Haluk Lacin for insightful comments on manuscript. We also acknowledge the Washington University Zebrafish Facility for fish care and Washington University Center for Cellular Imaging (WUCCI) for supporting the confocal imaging experiments. This work was supported by funding through the National Institute of Health (NIH) R01 DC016413 (MWB). MWB is a Pew Biomedical Scholar and a McKnight Foundation Scholar.

# Additional information

## Funding

| Funder | Grant reference number | Author |
|---|---|---|
| National Institute on Deafness and Other Communication Disorders | R01 DC016413 | Martha W Bagnall |
| Pew Charitable Trusts | | Martha W Bagnall |
| McKnight Endowment Fund for Neuroscience | | Martha W Bagnall |

The funders had no role in study design, data collection and interpretation, or the decision to submit the work for publication.

## Author contributions

Saul Bello-Rojas, Formal analysis, Investigation, Visualization, Methodology, Writing - original draft, Writing - review and editing; Martha W Bagnall, Conceptualization, Supervision, Funding acquisition, Writing - review and editing

## Author ORCIDs

Saul Bello-Rojas ⓘ http://orcid.org/0000-0002-8090-9577
Martha W Bagnall ⓘ http://orcid.org/0000-0003-2102-6165

## Ethics

All experiments and procedures were approved by the Animal Studies Committee at Washington University within IACUC protocol 20-0367 and adhere to NIH guidelines.

## Decision letter and Author response

Decision letter https://doi.org/10.7554/eLife.83680.sa1
Author response https://doi.org/10.7554/eLife.83680.sa2

# Additional files

## Supplementary files

• MDAR checklist

## Data availability

All analyses generated during this study are included in the manuscript source data file. Computational analyses were performed using code available at https://github.com/bagnall-lab/Event-detection, (copy archived at swh:1:rev:b2fbdb9c9acb263116c2f041caf89d28d1ed1c5f) (Bagnall, 2022). Raw data will be available on Dryad at the following DOI: https://doi.org/10.5061/dryad.4xgxd25dh.

The following dataset was generated:

| Author(s) | Year | Dataset title | Dataset URL | Database and Identifier |
|---|---|---|---|---|
| Bello Rojas S, Bagnall M | 2022 | Clonally related, Notch-differentiated spinal neurons integrate into distinct circuits EPhys raw data | https://doi.org/10.5061/dryad.4xgxd25dh | Dryad Digital Repository, 10.5061/dryad.4xgxd25dh |

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
