## [Editor Report]

This is an important article that describes the connectivity of sibling neurons in the zebrafish spinal cord, where one sibling receives Notch signaling (Notch-ON) and the other does not (Notch-OFF). They find that V2a and V2b siblings have different morphology, inputs, outputs, and fail to connect to each other; this provides new insight into the role of lineage in specifying neuronal connectivity. The experiments are convincing and the conclusions are supported by the data presented.

---

## [Decision Letter]

**Decision letter after peer review:**

Thank you for submitting your article "Clonally related, Notch-differentiated spinal neurons integrate into distinct circuits" for consideration by *eLife*. Your article has been reviewed by 3 peer reviewers, including Chris Q Doe as Reviewing Editor and Reviewer #1, and the evaluation has been overseen by Claude Desplan as the Senior Editor. The individuals involved in the review of your submission have agreed to reveal their identity: Bruce Appel (Reviewer #2); Kamal Sharma (Reviewer #3).

Essential revisions:

There are no essential revisions or further experiments. The authors are encouraged to address the requested text changes noted by reviewers 1 and 2 prior to uploading a final version. There will be no re-review.

*Reviewer #1 (Recommendations for the authors):*

1) Optional NOT required: co-express constitutively active Notch (or Notch knockdown constructs) in the vsx1+ siblings and determine if equalizing Notch levels generates similar morphology or connectivity. Right now the paper is strictly correlative, which is fine, but functional assays manipulating Notch activity would greatly strengthen the paper.

2) It would be nice to discuss where the asymmetry in Notch arises; is it due to asymmetric partitioning of Numb during the terminal mitosis like in the fly, or is it due to localization of a Notch ligand such that only one sibling contacts the ligand?

3) The description of how sibling neurons are identified is good in the legend to figure 2, but vague and confusing in the main text (page 6). Please clarify the text.

4) The title "Sister V2a/b neurons remain proximal to each other" does not use the term proximal correctly. I suggest replacing "proximal" with "adjacent" or "in proximity".

5) On Page 8, line 156 the term "positioned in different clusters" is unclear to me. Can you clarify?

6) Page 9 line 177 what about at fast swim speeds?

7) Page 10 line 210 most journals do not allow "data not shown" so either add it as a supplement or simply delete it.

8) Figure 7H and I please use white letters on dark backgrounds.

9) Figure 2A, the red arrow is hard to see; perhaps make it yellow?

*Reviewer #2 (Recommendations for the authors):*

1. "Notch-differentiated" needs some additional introduction and context. Some people who read this manuscript will not be familiar with the role of Notch in the neuronal specification.

2. Page 8, line 156: "neurons….are positioned in different clusters". This needs some additional explanation. Specifically, it is not clear what "clusters" means in this context.

3. Figure 3, panel A. In this image, neither vsx1:EGFP+ interneuron appears to express chx10:Red.

4. Figure 4, panel B. The sister neurons appear to be further apart than expected from the preceding descriptions. A scale bar would be helpful.

5. An explicit explanation of how the authors are certain that they tested V2a/b pairs for the experiments shown in Figures 6 and 7 would be helpful.

---

## [Author Response]

Reviewer #1 (Recommendations for the authors):1) Optional NOT required: co-express constitutively active Notch (or Notch knockdown constructs) in the vsx1+ siblings and determine if equalizing Notch levels generates similar morphology or connectivity. Right now the paper is strictly correlative, which is fine, but functional assays manipulating Notch activity would greatly strengthen the paper.

We agree that manipulations of Notch might alter connectivity of the circuit. Previous Notch manipulations in Mizoguchi et al., 2020 resulted in either an increase or decrease in V2a/b numbers, so we hypothesize that a temporally controlled Notch manipulation would be needed after V2a/b neurons become post-mitotic to determine whether Notch itself dictates connectivity, or simply neuronal identity. We now address this in the Discussion (Lines 274-279).

2) It would be nice to discuss where the asymmetry in Notch arises; is it due to asymmetric partitioning of Numb during the terminal mitosis like in the fly, or is it due to localization of a Notch ligand such that only one sibling contacts the ligand?

The asymmetry in Notch arises after the Vsx1 progenitor division where asymmetric expression of Δ ligands leads to upregulation of Notch levels in the V2b and downregulation in the V2a via subsequent lateral inhibition as discussed by Okigawa et al., 2014 and Francius et al., 2016. This is now addressed in the Introduction (Lines 40-44).

3) The description of how sibling neurons are identified is good in the legend to figure 2, but vague and confusing in the main text (page 6). Please clarify the text.

We have now clarified the description of how sibling and non-sibling V2a/b sister neurons were identified in the Results section of Figure 2 (lines 119-121).

4) The title "Sister V2a/b neurons remain proximal to each other" does not use the term proximal correctly. I suggest replacing "proximal" with "adjacent" or "in proximity".

Agreed. The Results section title (Line 113) was changed to “Sister V2a/b neurons remain in close proximity to each other.” In addition, sister V2a/b somata are now described as being “in close proximity to each other” throughout the manuscript and the figure legend (Page 24).

5) On Page 8, line 156 the term "positioned in different clusters" is unclear to me. Can you clarify?

We agree that our original terminology was unclear in reference to the different Notch^ON^/Notch^OFF^ hemilineages found in the *Drosophila*. We have reworded this to “In contrast, Notch^ON^ and Notch^OFF^ sister neurons in *Drosophila* integrate into separate hemilineages that segregate spatially, although whether they receive shared input is not known”.

6) Page 9 line 177 what about at fast swim speeds?

During our paired whole cell recordings, we were not always able to induce fictive swim at fast speeds, so we cannot make a conclusion on the synaptic input of sister V2a/b neurons at fast speeds. We address this by adding (Lines 181-183), “We cannot rule out the possibility that sister neurons receive shared inputs from circuits for fast locomotion or specialized behaviors.”

7) Page 10 line 210 most journals do not allow "data not shown" so either add it as a supplement or simply delete it.

We have removed this statement from the manuscript.

8) Figure 7H and I please use white letters on dark backgrounds.

Done.

9) Figure 2A, the red arrow is hard to see; perhaps make it yellow?

We tried yellow and found it was difficult to see with the white soma; instead, we have improved visibility by changing the arrow color to a dark red.

Reviewer #2 (Recommendations for the authors):1. "Notch-differentiated" needs some additional introduction and context. Some people who read this manuscript will not be familiar with the role of Notch in the neuronal specification.

We have now clarified the abstract by adding the sentence “In *Drosophila*, in contrast, sister neurons with different levels of Notch expression (Notch^ON^/Notch^OFF^) develop distinct identities and diverge into separate circuits” as well as by adding the Notch^ON/OFF^ terminology to the sister V2a/b neurons.

2. Page 8, line 156: "neurons….are positioned in different clusters". This needs some additional explanation. Specifically, it is not clear what "clusters" means in this context.

The term, clusters, was referring the different Notch^ON^/Notch^OFF^ hemilineages found in the *Drosophila*, each consisting of different groups/clusters of neurons. We have reworded this to “In contrast, Notch^ON^ and Notch^OFF^ sister neurons in *Drosophila* integrate into separate hemilineages that segregate spatially, although whether they receive shared input is not known” as detailed by Harris et al., 2015.

3. Figure 3, panel A. In this image, neither vsx1:EGFP+ interneuron appears to express chx10:Red.

We have substituted a different example image to include a *vsx1* GFP+ sister pair where *chx10:Red* is more visible (Page 25). In the earlier figure, one of the sister neurons did express *chx10:Red* but the expression was swamped by the dominance of the GFP label, leading to the apparent absence.

4. Figure 4, panel B. The sister neurons appear to be further apart than expected from the preceding descriptions. A scale bar would be helpful.

A scale bar (10 µm) was added to panel. These sister neurons are ~15 µm apart, consistent with typical values reported in Figure 2B.

5. An explicit explanation of how the authors are certain that they tested V2a/b pairs for the experiments shown in Figures 6 and 7 would be helpful.

Morphological assessments of V2a/b/s reveal that each group has some distinct characteristics as shown by Menelaou et al., 2014, Callahan et al., 2019, and Gerber et al., 2020. Particularly, V2a neurons in zebrafish have a distinct ascending branch that can extend one to several segments rostral. In contrast, V2b and V2s neurons only have descending projections. We used these key differences to identify Vsx1 sister pairs with a V2a neuron present. For better clarity, we have now addressed this explicitly in Results (lines 205-207) and Methods (392-396).